# Evaluation of the Antioxidant and Antimicrobial Activity of Natural Deep Eutectic Solvents (NADESs) Based on Primary and Specialized Plant Metabolites

**DOI:** 10.3390/molecules30214219

**Published:** 2025-10-29

**Authors:** Magdalena Kulinowska, Agnieszka Grzegorczyk, Sławomir Dresler, Agnieszka Skalska-Kamińska, Katarzyna Dubaj, Maciej Strzemski

**Affiliations:** 1Department of Analytical Chemistry, Medical University of Lublin, 20-093 Lublin, Poland; kulinowskamagdalena@gmail.com (M.K.); agnieszka.skalska-kaminska@umlub.edu.pl (A.S.-K.); 2Department of Pharmaceutical Microbiology, Medical University of Lublin, 20-093 Lublin, Poland; agnieszka.grzegorczyk@umlub.edu.pl; 3Department of Plant Physiology and Biophysics, Institute of Biological Sciences, Faculty of Biology and Biotechnology, Maria Curie-Skłodowska University, 19 Akademicka St., 20-033 Lublin, Poland; 4Department of Basic Medical Sciences, Faculty of Medical and Health Sciences, Casimir Pulaski Radom University, 27 Boleslawa Chrobrego Str., 26-600 Radom, Poland; k.dubaj@urad.edu.pl

**Keywords:** methicyllin-resistant *Staphylococcus aureus*, disc diffusion method, green chemistry, lipophilic NADES, flavonoids, phenolic acids, monoterpenes, bioautography, PCA, PLS-DA

## Abstract

NADESs represent a modern class of extraction media that align with the principles of green chemistry. They are considered non-toxic and biodegradable, but relatively little is known about their biological activity. This study investigated the antioxidant, antibacterial, and antifungal properties of 40 NADESs. The systems were developed from primary (PRIM) based on choline chloride (ChCl), and specialized (HEVO) plant-derived metabolites, particularly based on thymol and menthol. Their antioxidant activity was evaluated using spectrophotometric tests. The antimicrobial activity was evaluated by the disk diffusion method. The data obtained were analyzed using principal component analysis (PCA) and partial least squares discriminant analysis (PLS-DA). NADESs based on PRIM exhibited negligible antioxidant activity and relatively low antimicrobial activity. By contrast, NADESs containing HEVO, particularly thymol-based systems, indicated significant antioxidant activity, with stronger activity observed at higher molar proportions of thymol. In the 1,8-cineole:thymol system, ABTS activity ranged from 167.37 ± 24.17 to 861.25 ± 33.03 mg Trolox equivalents/mL NADES (molar ratios 9:1 and 1:9, respectively). The 1,8-cineole:thymol system (1:9) also showed strong antimicrobial activity, with a maximum inhibition zone of 39.33 ± 2.52 mm against *Staphylococcus aureus*. In summary, NADESs based on HEVO exhibit significantly stronger biological activity than those containing only PRIM.

## 1. Introduction

Natural deep eutectic solvents (NADESs) have become the subject of numerous studies in recent years, focusing on their application in the extraction of plant compounds, chemical synthesis and the formulation of pharmaceutical products. These solvents are compositions of natural substances, both biosynthetically primary metabolites (PRIM) such as quarternary amines, organic acids, amino acids, fatty acids, sugars, and polyols, and highly evolutionary metabolites (HEVO) such as alkaloids, flavonoids, monoterpenes, and phenols [1,2,3]. Common NADES combinations include choline chloride:glycerol, choline chloride:lactic acid, thymol:menthol, and camphor:thymol (typically at a 1:1 molar ratio) [2,4].

These mixtures melt at a temperature significantly lower than their individual components, forming liquid systems at room temperature and often even much lower. It is assumed that the use of NADESs complies with the principles of green chemistry, as they are generally considered non-toxic, biodegradable, and biocompatible [5,6,7]. In addition, some NADESs possess desirable biological properties—such as antioxidant, antibacterial, and antifungal activities [8,9,10,11]—which represent significant advantages over conventional solvents. These properties make extracts obtained using NADESs more durable—by eliminating the risk of oxidation of the extracted compounds and enhancing microbiological stability. Moreover, they exhibit biological activity, resulting from the synergistic interaction between NADESs and the extracted compounds [8]. Furthermore, investigating the antimicrobial activity of NADESs seems particularly promising in the context of the increasing antibiotic resistance of pathogenic microorganisms [11].

Unfortunately, despite the well-known activities of many substances forming NADES systems, there are few reports on the activities of eutectic solvents designed on their basis. The unique supramolecular structure formed through hydrogen bonding between NADES components may exhibit distinct physicochemical and biological properties compared to the properties of the individual compounds [12]. An additional difficulty is the fact that the number of possible NADES combinations is almost unlimited, and extensive research is required to fully characterize their biological activities. This is particularly important for the pharmaceutical and food industries, because NADESs are increasingly used as drug solvents and extractants [1,8]. Several studies have demonstrated that NADES formulations significantly enhance the extraction efficiency of plant metabolites, such as alkaloids [7], lichen metabolites [3], triterpenes [13], anthocyanins [14], isoflavones [15,16], chlorogenic acid from various plant sources, and curcuminoids from *Curcuma longa* L., due to the higher solubility of these bioactive compounds in NADESs [4]. Furthermore, these solvents can be used in food cryoprotection [17], removal of heavy metals from food products [18,19], and as flavor enhancers [20] and food films [21]. This also prompts in-depth research into the biological activity of NADESs.

This study attempts a comprehensive assessment of the antioxidant and antimicrobial activity of 40 NADES compositions based on both PRIM and HEVO. Nineteen hydrophilic NADESs (11 PRIM-based and 8 HEVO-based compositions) and 21 lipophilic HEVO compositions forming volatile NADESs (VNADESs) were evaluated against both pathogenic fungi and bacteria (Gram-positive and Gram-negative). The aim of the study was to verify the hypothesis that NADES compositions containing HEVO have stronger antioxidant and antimicrobial properties than NADESs containing only PRIM, and that lipophilic NADESs exhibit more favorable biological properties than hydrophilic NADESs. Furthermore, we consider that the selection of NADESs with favorable biological activity will enable the rational design of extraction media, ensuring not only efficient extraction but also the stability of the obtained extracts, and additive pharmacological effects when crude NADES-based extracts are applied.

## 2. Results and Discussion

Liquids that are stable at room temperature were obtained by mixing the NADES components and heating them in a water bath at 60 °C for 30 min. After one week, only the NADESs based on choline chloride (ChCl) and arabinose had crystallized. The composition of NADESs tested is presented in Table 1.

### 2.1. Antioxidant Activity of Testing NADESs

Screening tests for antioxidant activity revealed that HEVO-based NADESs exhibited significant activity, whereas PRIM-based liquids showed low activity (except for liquids containing malic and citric acids in the DPPH test) (see Appendix A). Spectrophotometric assay (DPPH, ABTS, FRAP, and CUPRAC) also revealed that the NADESs developed using HEVO exhibited significantly higher antioxidant activity than those based solely on PRIM (see Figure 1 and Appendix A). The tested HEVO-based liquids, both hydrophilic and lipophilic, exhibited strong antioxidant activity against the ABTS radical and the ability to reduce Cu^2+^ ions to Cu^+^ (CUPRAC test). However, hydrophilic NADESs containing flavonoids and phenolic acids exhibited significantly higher antioxidant activity against the DPPH radical and the ability to reduce Fe^3+^ to Fe^2+^ (FRAP test), whereas lipophilic liquids (VNADESs) exhibited significantly lower activity in these tests (Figure 1). It is widely believed that the ABTS test is more sensitive than the DPPH test, and can be used to assess the antioxidant properties of both hydrophilic and lipophilic compounds [22]. Therefore, it appears to be suitable for a comprehensive assessment of the antioxidant properties of NADESs of varying polarity and composition. On the other hand, the CUPRAC test, performed in a neutral or slightly alkaline environment, is appropriate for assessing the activity of phenols (thymol), whose activity in the acidic environment of the FRAP test may be limited. Thus, as a rule, higher values are obtained for the same plant substances in the ABTS test compared to DPPH [23,24], and very often in the CUPRAC test compared to FRAP [25,26].

As shown in Figure 1 and Appendix A, antioxidant activity was observed in formulations containing flavonoids such as quercetin (165, 127, 200 and 166 mg Trolox equivalents per mL NADES (mg TE) for ABTS, DPPH, CUPRAC, and FRAP tests, respectively) and rutin (68, 114, 171 and 75 mg TE for ABTS, DPPH, CUPRAC, and FRAP tests, respectively) while the activity of the formulation containing naringin was negligible. Among formulations containing phenolic acids, those based on protocatechuic (395, 155, 813 and 330 mg TE for ABTS, DPPH, CUPRAC, and FRAP tests, respectively) and gentisic acids (522, 548, 502 and 530 mg ETx for ABTS, DPPH, CUPRAC, and FRAP tests, respectively) exhibited the strongest antioxidant properties. In contrast, compositions containing resorcylic and p-hydroxybenzoic acids generally exhibited no such activity, with the exception of α-resorcylic acid in the ABTS test.

Lipophilic NADESs exhibited significant antioxidant activity only in formulations containing thymol, with stronger activity observed at higher molar proportions of thymol. For example, for the 1,8-cineole:thymol system, the activity in the ABTS test ranged from 167 to 861 mg TE (for molar ratios of 9:1 and 1:9, respectively), while for the benzyl alcohol:thymol system, it ranged from 234 to 848 mg TE (for molar ratios of 9:1 and 4:6, respectively). The observed effect can be explained by the high antioxidant activity of thymol, which has been reported in numerous studies [27,28,29,30,31].

It should be noted that the volume of the reaction mixture in spectrophotometric tests was disproportionately large compared to the volume of NADESs added. Furthermore, some NADESs with significant activity had to be prediluted (e.g., NADESs containing gentisic acid). Thus, the mixtures subjected to spectrophotometric measurements constituted only solutions of NADES components, lacking the supermolecular structure that defines NADESs. This should be considered as a significant limitation of this study. However, preliminary bioautography tests, which involved using undiluted NADESs and evaporating free radical solvents by drying the chromatography plates, suggest that NADESs exhibit strong antioxidant activity even when their supermolecular structure is intact.

### 2.2. Antimicrobial Activity of Testing NADESs

To evaluate the antimicrobial and antifungal activity of different NADES formulations, we compared the broad spectrums of lipophilic and hydrophilic systems. Our findings highlighted a distinct advantage of lipophilic NADESs (HEVO-based) compared to hydrophilic ones. The superior performance of thymol-rich systems—exceeding even vancomycin (30 μg) in terms of the zone of inhibition (ZOI) against Gram-positive bacteria—highlights thymol’s potential as a natural antimicrobial agent when incorporated into NADES matrices (Figure 2; Appendix A). Numerous studies have demonstrated the efficacy of pure thymol as an antimicrobial agent against Gram-positive bacteria (*S. aureus*, *Staphylococcus epidermidis*, *Enterococcus faecalis*, *Bacillus cereus*), Gram-negative bacteria (*Escherichia coli*, *Salmonella* Typhimurium*, Pseudomonas aeruginosa*), and yeast (*Candida albicans*) [32,33,34].

Thymol’s antibacterial mechanisms are multifaceted, involving both membrane-level and intracellular actions. Its lipophilicity enables interaction with bacterial lipid membranes, disrupting membrane pumps and key enzymes. Furthermore, thymol can bind to large intracellular macromolecules, such as DNA, causing structural damage [32].

Thymol-based NADESs exhibited the most potent antimicrobial activity against all tested strains (Figure 2), except *P. aeruginosa*, which showed the highest sensitivity to the benzyl alcohol:menthol system. However, among all strains evaluated, *P. aeruginosa* was the least sensitive overall. In contrast, Gram-positive species were markedly more susceptible to thymol, emphasizing the influence of bacterial cell wall structure on antimicrobial sensitivity.

Lipophilic NADES systems demonstrated strong inhibitory effects against both methicillin-resistant *S. aureus* (MRSA) and methicillin-sensitive *S. aureus* strains (MSSA) (Figure 2; Appendix A). Several mixtures nearly doubled the ZOI compared to the positive control (vancomycin, 30 µg). Effective combinations for MSSA included 1,8-cineole:thymol (1:9), benzyl alcohol:thymol (4:6), and borneol:thymol (1:2 and 3:7). For MRSA, camphor:thymol (3:7) and borneol:thymol (2:3) also produced enhanced activity, suggesting a heightened sensitivity of MRSA to these NADES formulations. These findings indicate the potential of such systems to combat antibiotic-resistant strains effectively.

The anti-MRSA activity of thymol was previously described by Yuan et al. [35], who demonstrated that co-treatment with thymol and vancomycin was more effective in eliminating MRSA biofilms in a mouse infection model than vancomycin monotherapy. The study showed that thymol inhibited biofilm formation and disrupted mature biofilms by reducing the production of polysaccharide intracellular adhesin and suppressing the release of extracellular DNA [35].

Interestingly, while thymol-based mixtures generally showed increased antimicrobial activity with higher thymol content, camphor:thymol NADESs displayed an inverse trend, suggesting complex interactions between components that influence overall efficacy. These findings contrast with studies on individual compounds, such as (±)-camphor and thymol, which reported minimum inhibitory concentrations (MICs) of 0.25 mg/mL for camphor and 0.007 mg/mL for thymol against *B. cereus* [34]. These values indicate that higher thymol content should enhance anti-*B. cereus* activity, which contrasts with the trend observed in our camphor:thymol systems.

This deviation suggests that the antimicrobial activity of thymol-based NADESs results from synergistic, additive, or antagonistic interactions between NADES components. Oxygenated terpenes such as camphor and 1,8-cineole have been shown to enhance the efficacy of phenolic compounds like eugenol in essential oil systems [36]. This suggests that thymol-based combinations may rely on a similar underlying mechanism.

In Gram-negative *S.* Typhimurium, thymol and benzyl alcohol mixtures exhibited similar inhibitory effects regardless of molar ratio, suggesting comparable antimicrobial potency of these two components against this pathogen (Figure 3; Appendix A).

Certain findings from this study suggest that the antimicrobial effect may be attributed solely to thymol, such as in borneol-based systems against *E. faecalis* and *P. aeruginosa*. This assumption is based on previous studies reporting no antimicrobial activity of pure borneol against these strains [37]. In another study, *S.* Typhimurium showed susceptibility to (+)-borneol only at a very high MIC (800 mg/mL), while thymol demonstrated an MIC of 0.003 mg/mL, further supporting the prediction that the observed inhibition is solely based on the presence of thymol alone [34].

Previous research suggests that the reduced or absent activity of menthol may be related to its lack of a phenolic ring, which plays a role in electron destabilization and membrane disruption [38]. In menthol-based NADES formulations, increasing menthol content in 1,8-cineole:menthol systems correlated with increased antimicrobial activity. In contrast, camphor:menthol mixtures showed a decrease in activity with increasing menthol content. Based on trends observed in binary systems against *S. epidermidis*, camphor demonstrated the highest activity, followed by menthol and 1,8-cineole.

The tested NADES formulations exhibited limited activity against Gram-negative bacteria (Figure 3; Appendix A), with inhibition zones smaller than those produced by the positive control, ciprofloxacin (5 μg). The outer membrane of Gram-negative bacteria, rich in lipopolysaccharides, provides substantial protection against hydrophobic agents [39], likely explaining their lower sensitivity to lipophilic NADESs.

Nevertheless, lipophilic NADESs (HEVO-based) consistently exhibited larger zones of inhibition than their hydrophilic counterparts (PRIM- and HEVO-based), suggesting greater efficacy. Benzyl alcohol emerged as particularly effective against *P. aeruginosa*. Mixtures such as benzyl alcohol:thymol and benzyl alcohol:menthol, especially at a 9:1 molar ratio, produced the strongest antimicrobial effects among NADESs tested—though still inferior to ciprofloxacin (Figure 3; Appendix A).

Some researchers have demonstrated that the weak organic acid drug N-acetyl-L-cysteine (NAC) can effectively eradicate *P. aeruginosa* biofilms. They proposed that at pH < pKa, NAC is able to penetrate the matrix barrier, diffuse into bacterial cells, and achieve complete eradication of the biofilm [40].

While the mechanisms of our NADES formulations may differ, these findings highlight the potential role of acidity in overcoming *P. aeruginosa* resistance. In our study, acid-based hydrophilic NADESs (NADESs 1–4 and 15–19) showed notable antimicrobial activity against both Gram-positive and Gram-negative bacteria, including *P. aeruginosa*. This activity may stem from their acidic nature, which could disrupt bacterial enzymes or compromise membrane integrity, as also reported by Bedair and Samir [10].

In contrast, sugar-, polyalcohol-, and urea-based NADESs, (PRIM-based) and flavonoid-based NADESs consistently exhibited minimal or no antimicrobial activity, indicating their function primarily as inert solvents or stabilizers. This confirms that the antimicrobial effects of these formulations are primarily driven by the bioactivity of specific components, rather than by the NADES matrix as an inert system. Notably, dehydration effects often associated with carbohydrate-based NADESs were not observed in any tested pathogen, suggesting that this mechanism may not be relevant under current experimental conditions.

Hydrophilic NADESs (PRIM- and HEVO-based) exhibited negligible antifungal activity (Figure 4; Appendix A). In contrast, lipophilic NADESs showed strong antifungal effects, suggesting potential as antifungal agents. Among all tested organisms, yeasts were the most sensitive, followed by Gram-positive, then Gram-negative bacteria. Thymol, similar to camphor, has demonstrated significant antifungal properties, particularly against *Candida* species [32].

It is worth noting that the antibacterial and antifungal activity of the tested NADESs was inversely correlated with their antioxidant activity (Appendix A). As is well-known, there are compounds with both antioxidant and antimicrobial activity, and under certain conditions, positive correlations between their content and antibacterial and antifungal activity have been observed [41]. However, the negative effect of antioxidant activity on antibacterial and antifungal activity observed in this study may be the result of very high concentrations of NADESs in the microorganism growth environment.

### 2.3. Classification of NADESs Based on Antioxidant and Antimicrobial Profiles

PCA was performed to visualize the relationships between antioxidant and antimicrobial activities of the tested NADESs (Figure 5). It was found that the first two principal components explained 47.1% of the total variance (PC1: 28.9%, PC2: 18.2%), allowing a clear two-dimensional separation of the solvent systems. The score plot revealed distinct clusters corresponding to the different groups of NADESs (flavonoid-based, menthol-based, organic acid-based, phenolic acid-based, polyol-based, PRIM–PRIM, and thymol-based). Colored ellipses representing 95% confidence intervals confirmed consistent grouping within each cluster.

The antioxidant properties of the tested NADESs were strongly correlated with each other (Appendix A), and the loading vectors indicated that antioxidant assays were primarily negatively associated with PC1 and positively with PC2, which positioned phenolic acid- and flavonoid-based NADESs close to these variables, reflecting their strong antioxidant capacity. In contrast, antimicrobial activities were positively related to both PC1 and PC2, particularly the inhibitory responses against *C. albicans* and *C. glabrata*, which facilitated the separation of menthol- and thymol-based NADESs towards the antimicrobial space. Polyol- and organic acid-based NADESs were positioned more centrally, consistent with their moderate activity across both antioxidant and antimicrobial properties.

The hierarchical-clustering heatmap corroborated the PCA results and revealed clear grouping patterns (Figure 6). Figure 6 provides a comprehensive summary of the relationships between the type of NADESs and their mean antioxidant and antimicrobial activities, serving as an integrative overview of the biological performance of all systems. Comparison of standardized antioxidant and antimicrobial activities of the tested NADESs, according to their hydrophilic/lipophilic character and composition, showed that the first group of lipophilic NADESs, mainly HEVO-based and thymol-type, exhibited higher antimicrobial activity and generally stronger antioxidant capacity. In contrast, polyol- and organic acid-based NADESs were positioned in clusters with lower or moderate activity across both antioxidant and antimicrobial assays, together with some menthol-based NADESs. This group also contained phenolic- and flavonoid-based NADESs, which were characterized by relatively high antioxidant activity.

To further explore the discrimination power of the measured variables, a PLS-DA analysis was performed based on antioxidant and antimicrobial activities of the tested NADESs (Figure 7A). In contrast to the PCA, where seven groups were distinguished, the PLS-DA reduced the separation to three major groups according to their composition and polarity: Hydro/HEVO, Hydro/Prim, and Lipo/HEVO. The first two components explained together 42.1% of the variance (Component 1: 26.6%, Component 2: 15.5%), allowing for a clear visualization of the dataset. The cross-validation results demonstrated good model performance (Figure 7C). For one latent component, the model reached Accuracy = 0.71, R^2^ = 0.69, and Q^2^ = 0.54. With two components, classification markedly improved (Accuracy = 0.82, R^2^ = 0.72, Q^2^ = 0.56). For models with three and four components, although R^2^ slightly increased (0.74 and 0.76, respectively), Q^2^ values declined (0.51 and 0.36), suggesting a risk of overfitting. Thus, the model with two latent components was considered optimal, balancing high explanatory and predictive ability. The variable importance in projection (VIP) scores (Figure 7B) showed that among antimicrobial responses, inhibitory activities against *Candida* species played a dominant role in separating NADESs. Yeast inhibition was a key discriminator of Lipo/HEVO NADESs, which exhibited the highest activity against these microorganisms, as well as significant inhibitory effects against *E. coli* and *E. faecalis*. This observation was confirmed by the score plot (Figure 7A), which showed that Hydro/Prim NADESs and some Hydro/HEVO NADES clustered mainly along the negative values of Component 1, whereas the majority of Hydro/HEVO samples formed a distinct group on the positive side of Component 1. This component was largely influenced by inhibitory activity against *Candida* species, as well as activity against MSSA.

## 3. Materials and Methods

### 3.1. Materials and Chemicals

D-arabinose (≥99.0%), benzyl alcohol (≥99%), (-)-borneol (97%), (±)-camphor (≥95.5%), choline chloride (≥98%), 1,8-cineole (for synthesis), citric acid (≥99.5%), D-fructose (≥99%), gentisic acid (≥98%), D-(+)-glucose (≥99.5%), glycerol (≥99.0%), *p*-hydroxybenzoic acid (99%), DL-lactic acid (~90%), levulinic acid (98%), DL-malic acid (≥99%), DL-menthol (≥95%), naringin (≥95%), 1,3-propanediol (98%), protocatechuic acid (≥97%), quercetin (≥95%), α-resorcylic acid (≥97%), β-resorcylic acid (≥97%), rutin hydrate (≥94%), D-tagatose (≥98%), thymol (≥99%), urea (99.0-100.5%), DPPH, ABTS (98%), potassium persulfate (≥99%), 1,3,5-Tri(2-pyridyl)-2,4,6-triazine (TPTZ), iron (III) chloride (≥98%, FeCl_3_), sodium acetate (≥99%), copper(II) chloride (99%), neocuproine (≥98%), ammonium acetate (≥98%), and 6-hydroxy-2,5,7,8-tetramethylchroman-2-carboxylic acid (Trolox) were purchased from Sigma-Aldrich (St. Louis, MO, USA).

Acetic acid (99.5–99.9%) and hydrochloric acid (36%) were supplied by Avantor Performance Materials Poland S.A. (Gliwice, Poland). Water was deionized and purified using ULTRAPURE Milipore Direct-Q^®^ 3UV–R (Merck, Darmstadt, Germany). HPTLC RP-18WF254s plates were purchased from Merck (Darmstadt, Germany). All NADESs were stored in sealed amber glass vials at 22 ± 2 °C and under relative humidity below 40% until analysis.

### 3.2. Antioxidant Panel

The antioxidant activity of NADESs was screened using bioautography on HPTLC plates. Two microlitres of each liquid were applied to the adsorbent surface, after which the plate surface was sprayed, separately, with DPPH and ABTS free radical solutions. The plates were then photographed.

Quantitative antioxidant activity was evaluated using four complementary assays—DPPH [42], ABTS [43], FRAP, and CUPRAC [26]—providing a comprehensive assessment encompassing free radical scavenging (ABTS; 2,2′-azinobis-(3-ethylbenzthiazoline-6-sulfonic acid) and DPPH; 2,2-diphenyl-1-picrylhydrazyl) as well as Fe^3+^ (FRAP; Ferric Reducing Antioxidant Power) and Cu^2+^ (CUPRAC; Cupric ion Reducing Antioxidant Capacity). All measurements were performed using a UV–Vis Evolution 500 spectrophotometer (Thermo Electron Corporation, Waltham, MA, USA) and semi-micro cuvettes (Brand GMBH, Wertheim, Germany). HEVO-based NADESs, exhibiting particularly strong antioxidant activity, were diluted with methanol. For all tests, the NADES samples were incubated with the appropriate reagents for 30 min prior to measuring the absorbance. Values were expressed as the mg Trolox equivalent per mL NADES.

### 3.3. Antimicrobial Susceptibility Testing by Disk Diffusion Method (According to Kirby–Bauer)

#### 3.3.1. Microorganisms

The antimicrobial activity of the tested NADESs were evaluated in accordance with EUCAST recommendations [44] against representative Gram-positive bacteria strains (*S. aureus* ATCC 29213, *S. aureus* ATCC BAA-1707, *S. epidermidis* ATCC 12228, *B. cereus* ATCC 10876, *E. faecalis* ATCC 51299), Gram-negative bacteria (*S.* Typhimurium ATCC 14028, *E. coli* ATCC 25922, *P. aeruginosa* ATCC 27853), and according to CLSI recommendations [45] for yeasts: *C. albicans* ATCC 10231, *C. glabrata* ATCC 90030, *C. auris* CDC B11903. The selected bacterial and yeast strains represent clinically relevant and well-characterized reference strains widely used for antimicrobial susceptibility testing and antimicrobial research [46,47,48,49,50,51].

The strains were obtained from certified microorganism collections of the American Type Culture Collection (ATCC, Manassas, VA, USA) and the Centers for Disease Control and Prevention (CDC, Atlanta, GA, USA) and stored at 4 °C. Prior to testing, all strains were seeded on appropriate media to verify their viability and purity.

#### 3.3.2. Procedure

1.Preparation of inoculum: Bacterial and yeast suspensions were prepared in sterile 0.85% NaCl and adjusted to a 0.5 McFarland standard (approximately 1–2 × 10^8^ CFU/mL for bacteria, 1–5 × 10^6^ CFU/mL for yeasts).2.Plate inoculation: Microbiological media were poured aseptically into Petri dishes to a uniform height (approximately 4 mm) and left to dry at room temperature before inoculation. Mueller–Hinton agar (MHA; Sigma-Aldrich) for bacteria and Mueller–Hinton agar with 2% (*w*/*v*) glucose (MHA + 2% Glu) for yeast were inoculated evenly in three directions with the prepared microbial suspension, using a sterile swab to obtain uniform growth.3.Preparation and application of test disks: 20 µL of each NADES was applied to sterile, blank 6 mm diameter paper disks (Oxoid™ Antimicrobial Susceptibility Test Discs, Thermo Fisher Scientific, Basingstoke, Hampshire, UK) and left to absorb under sterile conditions.

Commercial antibiotic disks were used as positive controls for test validation: ciprofloxacin (5 µg; Becton, Dickinson and Company, Franklin Lakes, NJ, USA) for Gram-negative bacteria, vancomycin (30 µg; Oxoid™ Thermo Fisher Scientific, UK) for Gram-positive bacteria, and fluconazole (25 µg; BioMaxima, Lublin, Poland) for yeast.

The disks were placed on the surface of the inoculated agar using sterile tweezers. To ensure accurate and non-overlapping zones of inhibition, four disks were evenly spaced per plate for NADESs marked 1 to 19. For volatile mixtures (from 20 to 40), each disk was placed individually on a separate plate to prevent cross-interference.

Each test was performed in triplicate to ensure repeatability of results, and mean values were calculated.

4.Incubation: Plates were incubated at 35 ± 2 °C for 18 h for bacteria and 24 h for yeasts.5.Measurement of inhibition zones: After incubation, the diameters of the microbial growth inhibition zones (including the diameter of the disk) were measured and the result was given in millimeters.

For the tested mixtures, relative activity was assessed based on the size of the inhibition zones compared to positive controls.

##### Aseptic Conditions

All procedures were performed under aseptic conditions to prevent contamination. All materials and instruments were sterilized prior to use. Disks were handled exclusively with sterile forceps, and culture plates were stored and used under aseptic conditions.

All experiments involving pathogenic microorganisms were conducted in accordance with institutional biosafety regulations, in a biosafety level 3 (BSL-3) laboratory at the Department of Pharmaceutical Microbiology, Medical University of Lublin, under a biosafety protocol approved by the Institutional Biosafety Committee.

### 3.4. Statistical and Multivariate Analysis

In order to detect differences between NADESs, statistical analyses were carried out using the Kruskal–Wallis ANOVA followed by the Conover–Inman post hoc test or Welch’s ANOVA followed by the Games–Howell post hoc test in PQStat Software (PQStat v.1.8.6.126, Poznań, Poland). Multivariate analyses were performed in MetaboAnalyst 5.0 [52]. PCA, PLS-DA, and heat mapping with hierarchical clustering (Euclidean distance, Ward’s linkage method) were applied to evaluate the relationships between antioxidant and antimicrobial activities of NADESs. Prior to multivariate analysis, data were normalized by sum and log transformation. For PLS-DA models, 5-fold cross-validation was applied to assess model performance.

## 4. Conclusions

In summary, these findings reinforce the potential of lipophilic NADES systems, particularly those based on thymol, exhibited potent antioxidant, and antimicrobial activity against a broad spectrum of pathogens. The variability in efficacy across different binary systems highlights the importance of component interactions, which may enhance or suppress antimicrobial effects. These findings highlight the potential of NADES preparations as versatile, natural antibacterial agents that can be used in pharmaceuticals and food preservation. However, further research is needed to clarify the mechanisms of their antibacterial action, optimize the preparations for maximum efficacy, and evaluate their safety profile through cytotoxicity studies. In addition, to fully exploit the potential of NADESs in combating microbial infections, it will be necessary to study a broader spectrum of microorganisms and perform quantitative antibacterial tests, such as minimum inhibitory concentration (MIC), minimum bactericidal concentration (MBC), and minimum fungicidal concentration (MFC), as the disk diffusion method has limited precision in evaluating compounds with limited diffusion in agar (e.g., lipophilic NADESs) and does not provide data on MIC, MBC, and MFC values.

Furthermore, although trends suggest potential synergistic interactions, formal synergy tests would be necessary to confirm this.

## Figures and Tables

**Figure 1 molecules-30-04219-f001:**
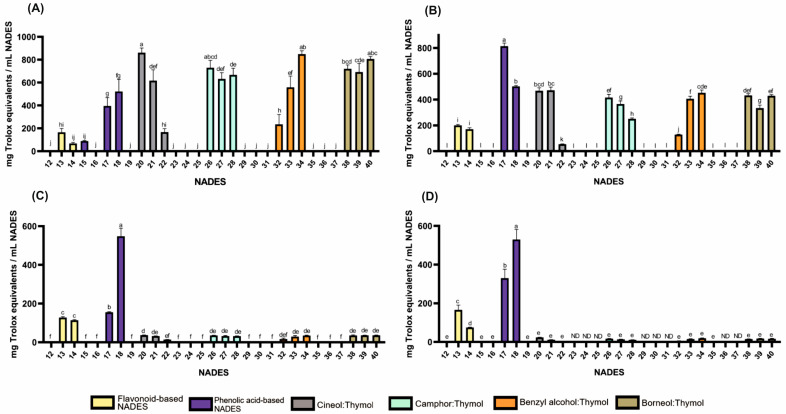
Antioxidant activities of HEVO-based NADESs (12–40) determined by assays: (**A**)—ABTS, (**B**)—CUPRAC, (**C**)—DPPH, and (**D**)—FRAP. NADES numbering is in accordance with Table 1. Columns marked with different lowercase letters represent significant differences based on Welch’s ANOVA followed by the Games–Howell post hoc test (*p* < 0.05). Data are presented as means (*n* = 3) ± standard deviation (SD). ND—not detected.

**Figure 2 molecules-30-04219-f002:**
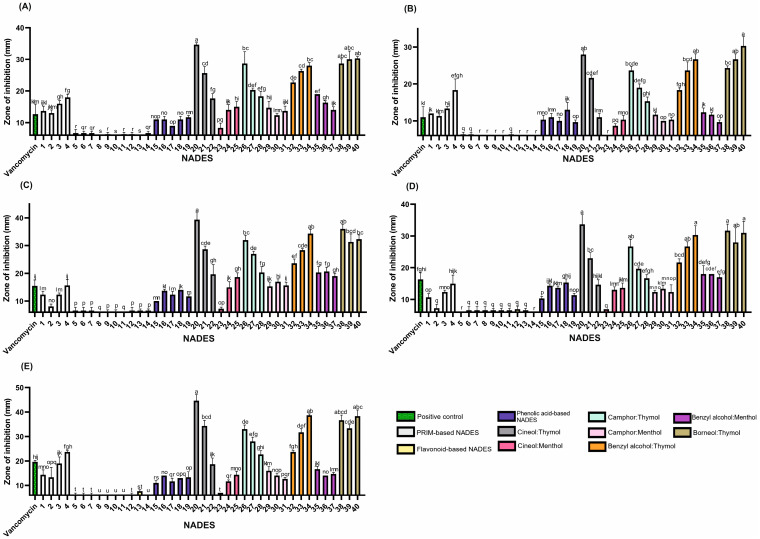
The antibacterial activity of the tested NADESs against selected Gram-positive strains: (**A**)—*B. cereus*, (**B**)—*E. faecalis*, (**C**)—MRSA, (**D**)—MSSA, and (**E**)—*S. epidermidis*. NADES numbering is in accordance with Table 1. Columns marked with different lowercase letters represent significant differences based on Kruskal–Wallis ANOVA followed by the Conover–Inman post hoc test (*p* < 0.05). Data are presented as means (*n* = 3) ± standard deviation (SD).

**Figure 3 molecules-30-04219-f003:**
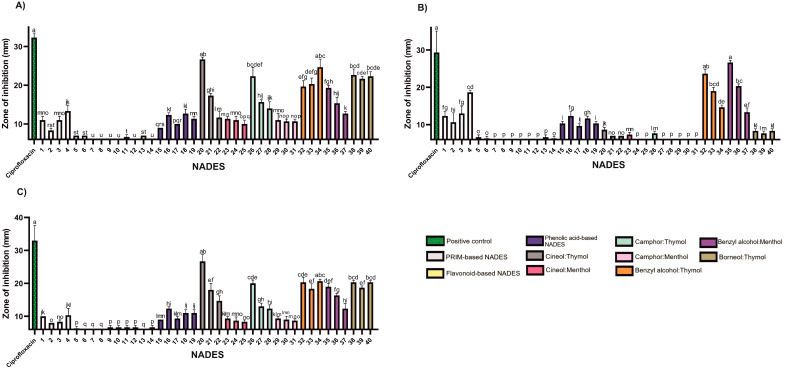
The antibacterial activity of the tested NADESs against selected Gram-negative strains: (**A**)—*E. coli*, (**B**)—*P. aeruginosa*, (**C**)—*S.* Typhimurium. NADES numbering is in accordance with Table 1. Columns marked with different lowercase letters represent significant differences based on Kruskal–Wallis ANOVA followed by the Conover–Inman post hoc test (*p* < 0.05). Data are presented as means (*n* = 3) ± standard deviation (SD).

**Figure 4 molecules-30-04219-f004:**
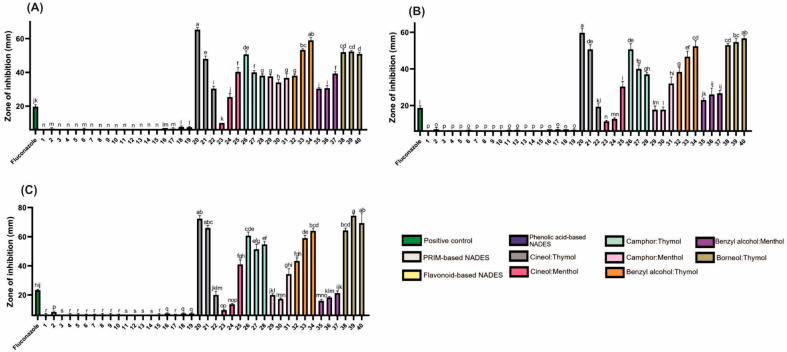
The antifungal activity of the tested NADESs against selected *Candida* strains: (**A**)—*C. albicans*, (**B**)—*C. glabrata*, (**C**)—*C. auris*. NADES numbering is in accordance with Table 1. Columns marked with different lowercase letters represent significant differences based on Kruskal–Wallis ANOVA, followed by the Conover–Inman post hoc test (*p* < 0.05). Data are presented as means (*n* = 3) ± standard deviation (SD).

**Figure 5 molecules-30-04219-f005:**
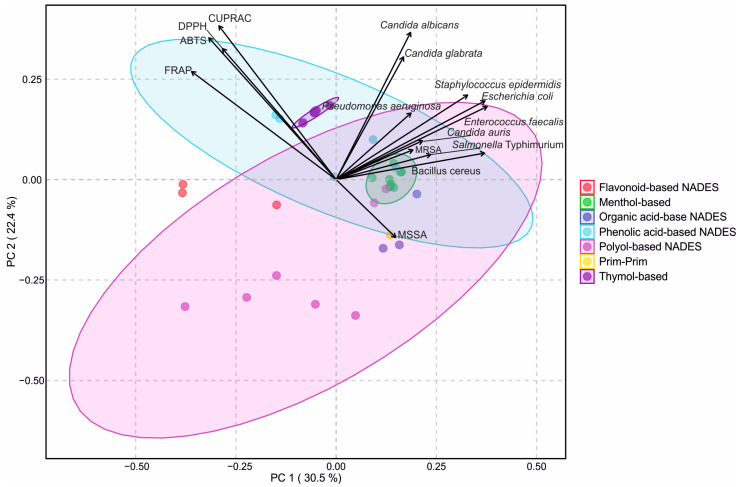
Principal component analysis (PCA) biplot of antioxidant and antimicrobial activities of natural deep eutectic solvents (NADES). Seven different NADES groups are shown: flavonoid-based, menthol-based, organic acid-based, phenolic acid-based, polyol-based, prim–prim, and thymol-based. Vectors represent antioxidant assays (ABTS, CUPRAC, DPPH, FRAP) and microbial strains. Colored ellipses indicate 95% confidence intervals for sample clustering within each NADES group.

**Figure 6 molecules-30-04219-f006:**
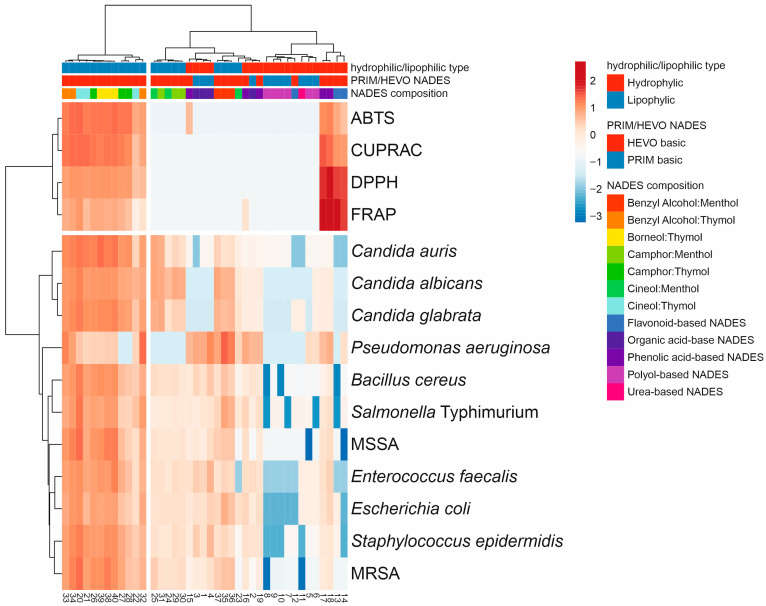
Comprehensive summary of the relationships between the type of NADESs and their mean antioxidant and antimicrobial activities as a visualization of the heat map of NADES biological activity diversity (numbering according to Table 1). The colors range from dark blue (low activity) to dark red (high activity). The hierarchical cluster dendrograms were constructed based on lipophilicity, composition, and biological properties of the tested NADESs.

**Figure 7 molecules-30-04219-f007:**
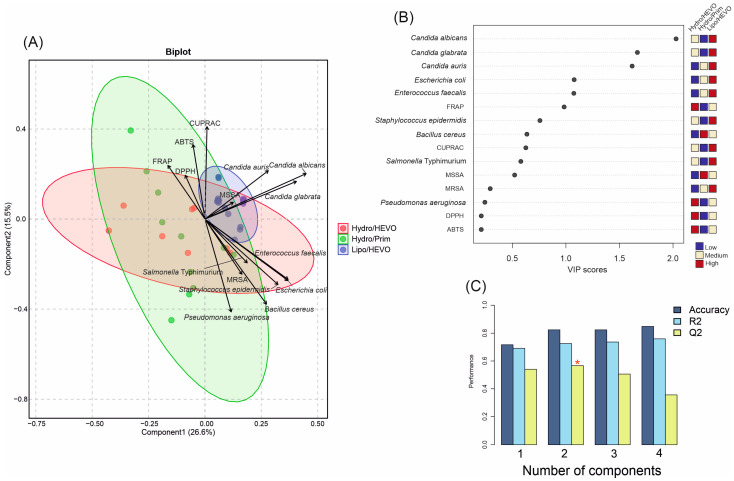
(**A**) Partial least squares discriminant analysis (PLS-DA) biplot of antioxidant and antimicrobial activities of NADESs belonging to three major groups (Hydro/HEVO, Hydro/Prim, and Lipo/HEVO). Vectors indicate the contribution of variables (antioxidant assays and microbial strains). Colored ellipses represent 95% confidence intervals illustrating sample clustering within each group. (**B**) Variable importance in projection (VIP) scores of PLS-DA highlighting the most discriminant parameters. (**C**) Cross-validation results of the PLS-DA model. The red asterisk denotes the optimal number of components determined by cross-validation.

**Table 1 molecules-30-04219-t001:** Qualitative and quantitative composition of the obtained NADESs.

No.	Component 1	Component 2	Component 3	Molar Ratio
Hydrophilic NADESs—PRIM-based				
1	ChCl	DL-lactic acid	water	2:1:2
2	ChCl	levulinic acid	water	2:1:2
3	ChCl	DL-malic acid	water	2:1:2
4	ChCl	citric acid	water	2:1:2
5	ChCl	1,3-propanediol	water	1:1:1
6	ChCl	glycerol	water	1:1:1
7	ChCl	D-(+)-glucose	water	2:1:6
8	ChCl	D-arabinose	water	2:1:6
9	ChCl	D-tagatose	water	2:1:6
10	ChCl	D-fructose	water	2:1:6
11	ChCl	urea	water	1:1:1
Hydrophilic NADESs—HEVO-based				
12	ChCl	naringin	water	2:0.1:3
13	ChCl	quercetin	water	2:0.1:3
14	ChCl	rutin hydrate	water	2:0.1:3
15	ChCl	α-resorcylic acid	water	4:1:4
16	ChCl	β-resorcylic acid	water	4:1:4
17	ChCl	protocatechuic acid	water	4:1:4
18	ChCl	gentisic acid	water	4:1:4
19	ChCl	*p*-hydroxybenzoic acid	water	4:1:4
Lipophilic NADESs—HEVO-based				
20	1,8-cineole	thymol	-	1:9
21	1,8-cineole	thymol	-	5:5
22	1,8-cineole	thymol	-	9:1
23	1,8-cineole	DL-menthol	-	9:1
24	1,8-cineole	DL-menthol	-	5:5
25	1,8-cineole	DL-menthol	-	1:9
26	(±)-camphor	thymol	-	3:7
27	(±)-camphor	thymol	-	5:5
28	(±)-camphor	thymol	-	6:4
29	(±)-camphor	DL-menthol	-	4:6
30	(±)-camphor	DL-menthol	-	3:7
31	(±)-camphor	DL-menthol	-	1:9
32	benzyl alcohol	thymol	-	9:1
33	benzyl alcohol	thymol	-	6:4
34	benzyl alcohol	thymol	-	4:6
35	benzyl alcohol	DL-menthol	-	9:1
36	benzyl alcohol	DL-menthol	-	6:4
37	benzyl alcohol	DL-menthol	-	4:6
38	(−)-borneol	thymol	-	2:3
39	(−)-borneol	thymol	-	1:2
40	(−)-borneol	thymol	-	3:7

## Data Availability

The data presented in this study are available on request from the corresponding author.

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
