# Peer review of "Evaluation of the Antioxidant and Antimicrobial Activity of Natural Deep Eutectic Solvents (NADESs) Based on Primary and Specialized Plant Metabolites"

_molecules, 2025, doi:10.3390/molecules30214219_

Round 1
Reviewer 1 Report
Comments and Suggestions for Authors
I have carefully reviewed the manuscript title “ Evaluation of the Antioxidant and Antimicrobial Activity of 2 Natural Deep Eutectic Solvents (NADESs) Based on Primary 3 (PRIM) and Specialized (HEVO) Plant Metabolites” The manuscript is very interesting and covers relatively newer aspect in plants metabolite science. I have the following suggestions;
- “NADESs are a modern class of extraction and reaction media a align with the principles of 18 green chemistry……!” It seems to have a grammar mistake.
- I would suggest to add key findings of all major experiments in abstract and Also add ±SD values with inhibition zones.
- Bacterial strains and reason for using specific bacteria must be mentioned
- Line 33-34 “Given the high diversity of NADES' biolog- 33 ical properties, this should be considered when selecting an extraction medium…” what should be considered??
- I would suggest to re write Abstract and pay attention on overall English grammar of manuscript.
- In introduction authors should Add key examples of NADES systems.
- Line 73 to 80 is unnecessary and placed at wrong section
- Most of used compounds have already significant antioxidant and antimicrobial potential so I am not clear regarding Novelity. Please clarify
Author Response
We would like to express our sincere gratitude to all reviewers for their valuable time, insightful comments, and constructive suggestions. We truly appreciate the effort invested in the careful evaluation of our manuscript. The feedback provided has been extremely helpful in improving the clarity, scientific rigor, and overall quality of the work. We have carefully addressed all comments point by point, and detailed responses, including the corresponding revisions made to the manuscript, are provided below.
Question 1
“NADESs are a modern class of extraction and reaction media a align with the principles of green chemistry…..!” It seems to have a grammar mistake.
Answer:
We thank the Reviewer for that thoughtful comment. Yes, definitely it is grammar mistake. We have changed sentence in Abstract section to “NADESs represent a modern class of extraction media that align with the principles of green chemistry.”
Question 2
I would suggest to add key findings of all major experiments in abstract and Also add ±SD values with inhibition zones.
Answer:
We greatly appreciate the Reviewer's very crucial suggestions. As the rewiever recommended, we have added some key findings of antioxidant and antimicrobial activity of NADESs in abstract. We have detailed the conclusions obtained by providing examples of specific values “By contrast, NADESs containing HEVO, particularly thymol-based systems, indicated significant antioxidant activity, with stronger activity observed at higher molar pro-portions of thymol. In the 1,8-cineole:thymol system, ABTS activity ranged from 167.37±24.17 to 861.25±33.03 mg Trolox equivalents/mL NADES (molar ratios 9:1 and 1:9, respectively). The 1,8-cineole:thymol system (1:9) also showed strong antimicrobial activity, with a maximum inhibition zone of 39.33±2.52 mm against MRSA (Staphylo-coccus aureus).”
Question 3
Bacterial strains and reason for using specific bacteria must be mentioned
Answer:
We thank the Reviewer for this suggestion and we strongly agree with it. As we selected broad range of gram(+), gram (-) bacteria and Candida spp. we added appropriate references which prove reason of our choice. We added sentence in Microorganisms subsection 3.3.1. (Materials and Methods section): “The selected bacterial and yeast strains represent clinically relevant and well-characterized reference strains widely used for antimicrobial susceptibility testing and antimicrobial research.”
Question 4
Line 33-34 “Given the high diversity of NADES’ biological properties, this should be considered when selecting an extraction medium…” what should be considered??
Answer:
We thank the Reviewer for the valuable suggestion. Due to the strict word limit for the abstract, the sentence was removed to maintain conciseness. However, we hope that the concluding sentence in the Introduction section “[…] we consider that the selection of NADESs with favorable biological activity will enable the rational design of extraction media, ensuring not only efficient extraction but also the stability of the obtained extracts, and additive pharmacological effects when raw crude NADES-based extracts are applied.” clearly addresses the highlighted aspect.
We believe this revised wording better reflects our intention, namely that the choice of extraction solvent should not be based solely on its extraction efficiency toward the target metabolites, but may also take into account the additional biological activity of the solvent.
Question 5
I would suggest to re write Abstract and pay attention on overall English grammar of manuscript.
Answer:
Thank you for drawing our attention to the language issues.
Depending on the previous reviewer’s suggestions abstract is rewrite and hopefully it should meet all requirements. We carefully re-read the entire manuscript and corrected all grammatical and stylistic errors we were able to identify.
Question 6
In introduction authors should Add key examples of NADES systems.
Answer:
We thank the reviewer for this suggestion. We revised the general statement referring to the studied NADES to a formulation that more explicitly specifies the investigated systems. The updated version now reads:
“Common NADES combinations include choline chloride:glycerol, choline chlo-ride:lactic acid, thymol:menthol, and camphor:thymol (typically at a 1:1 molar ratio).” We believe this revised wording provides a clearer and more informative description of the compositions evaluated in this study.
Question 7
Line 73 to 80 is unnecessary and place at wrong section
Answer:
We thank the Reviewer and really appreciate thorough insight into our manuscript. As recommended, we have shortened this sentence in Introduction and moved the abbreviations of the tests used to the Materials and Methods section, subsection 3.2. Antioxidant Panel.
Question 8
Most of used compounds have already significant antioxidant and antimicrobial potential so I am not clear regarding Novelity. Please clarify
Answer:
We appreciate the Reviewer’s insightful comment. The novelty of our research lies in the observation of trends in NADES systems, which differ significantly from ideal solutions of pure compounds. Our study was designed as a preliminary investigation aimed at identifying general trends and indicating which NADES formulations might serve as promising candidates for further development as antioxidant or antimicrobial extraction media.
As emphasized in the Introduction: “Unfortunately, despite the well-known activities of many substances forming NADES systems, there are few reports on the activities of eutectic solvents designed on their basis. The fact that hydrogen bonds form between NADES components, creating a new supramolecular structure, means that these solvents can exhibit activities distinct from those of their individual components [11].”
Furthermore, in the Results and Discussion section, we highlight that although the antimicrobial properties of pure thymol are well documented, the variability observed across different binary NADES systems indicates that component interactions can significantly influence the overall biological activity. We have also underscored in the Conclusion the importance of further research to better elucidate the mechanisms underlying these effects. We acknowledge the potential for synergistic interactions and explicitly state in the manuscript: “Although trends suggest potential synergistic interactions, formal synergy tests would be necessary to confirm this.”
If deemed necessary by the reviewers, we are willing to clarify this rationale more explicitly in the revised manuscript to better reflect the preliminary nature of our findings and the need for further targeted studies.
Reviewer 2 Report
Comments and Suggestions for Authors
It would be highly important for your manuscript to include a summary figure that correlates the type of NADES with the average antioxidant and antimicrobial activities.
It would also be valuable to provide the temperature and humidity parameters used during the storage of the NADES, as well as the model files (PCA/PLS-DA) to allow replication of the statistical analyses.
The purity level of all components and the moisture control procedures are not described and should be detailed.
The discussion is solid and well-supported by appropriate references; however, it is recommended to include an analysis of the mechanisms underlying NADES–membrane interactions.
It is further suggested to incorporate a pharmacological and toxicological profile, as this information is essential for potential pharmaceutical applications.
Author Response
We would like to express our sincere gratitude to all reviewers for their valuable time, insightful comments, and constructive suggestions. We truly appreciate the effort invested in the careful evaluation of our manuscript. The feedback provided has been extremely helpful in improving the clarity, scientific rigor, and overall quality of the work. We have carefully addressed all comments point by point, and detailed responses, including the corresponding revisions made to the manuscript, are provided below.
Question 1
It would be highly important for your manuscript to include a summary figure that correlates the type of NADES with the average antioxidant and antimicrobial activities.
Answer:
We thank the Reviewer for this thoughtful suggestion. We would like to emphasize that Figure 6 (heatmap visualization with hierarchical clustering) already serves as a summary figure correlating the type of NADES with their average antioxidant and antimicrobial activities. This figure integrates both types of biological responses and clearly groups NADES according to their composition and polarity (PRIM-based, HEVO-based, hydrophilic, and lipophilic). To make this clearer, we have added an explanatory sentence in the Results section and updated the figure caption accordingly.
Question 2
It would also be valuable to provide the temperature and humidity parameters used during the storage of the NADES, as well as the model files (PCA/PLS-DA) to allow replication of the statistical analyses.
Answer:
We thank the Reviewer for this valuable suggestion. The storage conditions of all NADES formulations have been specified in the revised manuscript (Section 3.1, Materials and Methods):
“All NADESs were stored in sealed amber glass vials at 22 ± 2 °C and under relative humidity below 40% until analysis.”
In addition, to ensure full reproducibility of our multivariate statistical analyses, all datasets and model files used for PCA and PLS-DA have been included as Supplementary Materials in a compressed ZIP archive.
Question 3
The purity level of all components and the moisture control procedures are not described and should be detailed.
Answer:
The purity levels of all components used in this study were provided in the Materials section. All chemicals were of analytical grade or higher, as specified in the text (e.g., ≥98% purity), and were used without further purification.
Regarding moisture control, no additional analytical procedures were performed to determine water content. However, all hygroscopic reagents were stored in tightly sealed containers under ambient laboratory conditions to minimize moisture uptake. These standard handling procedures were consistently applied throughout the study.
If the Reviewer or Editor considers it necessary, we would be happy to include this clarification in the revised manuscript.
Question 4
The discussion is solid and well-supported by appropriate references; however, it is recommended to include an analysis of the mechanisms underlying NADES-membrane interactions.
Answer:
We thank the Reviewer for this valuable comment. In the manuscript, we have indicated that “Thymol’s antibacterial mechanisms are multifaceted, involving both membrane-level and intracellular actions. Its lipophilicity enables interaction with bacterial lipid membranes, disrupting membrane pumps and key enzymes,” suggesting that similar mechanisms might underlie the activity of thymol-based NADES. However, we would like to clarify that a detailed mechanistic investigation of NADES–membrane interactions is beyond the scope of the present study, which focuses primarily on the antimicrobial evaluation of NADES. We agree that this is an important area for future research. We agree that this is an important area for future research. As noted in the Conclusion section (Section 4), we have already addressed this point by stating: “However, further research is needed to clarify the mechanisms of their antibacterial action, optimize the preparations for maximum efficacy, and evaluate their safety profile through cytotoxicity studies.”
Question 5
It is further suggested to incorporate a pharmacological and toxicological profile, as this information is essential for potential pharmaceutical applications.
Answer:
We fully agree that the pharmacological and toxicological profile is essential when considering potential pharmaceutical applications. This aspect was acknowledged in the Introduction, where we state that “It is assumed that the use of NADES complies with the principles of green chemistry, as they are generally considered non-toxic, biodegradable and biocompatible.” Moreover, in the Conclusion (Section 4), we emphasized the need for further studies in this area by stating that “further research is needed to […] evaluate their safety profile through cytotoxicity studies.” We believe these statements collectively reflect our awareness of the importance of this topic and indicate directions for future research.
Reviewer 3 Report
Comments and Suggestions for Authors
If natural deep eutectic solvents (NADES) are phytochemical extraction performance enhancers for the development of pharmaceutical and nutraceutical products, where is this development in research? To which plant was it applied, and what type of target metabolite was sought in the plant?
Why do they talk about antioxidant and antimicrobial activity in the title? Why do they talk about antioxidant and antimicrobial activity in the title? The manuscript also includes antifungal activity.
Why test the antioxidant activity of phenolic compounds that are known to have this property, as stated on line 130 of page 4? The observed effect can be explained by the high antioxidant activity of thymol, which has been reported in numerous studies [22–26].
Why test the antimicrobial activity of compounds that are known to have that property, as stated on line 155 of page 7? Numerous studies have demonstrated the efficacy of pure thymol as an antimicrobial agent against Gram-positive bacteria (S. aureus, S. epidermidis, E. faecalis, B. cereus), Gram-negative bacteria (E. coli, S. Typhimurium, P. aeruginosa), and yeast (C. albicans) [27–29].
I think I should consider giving another twist to the research presented.Author Response
We would like to express our sincere gratitude to all reviewers for their valuable time, insightful comments, and constructive suggestions. We truly appreciate the effort invested in the careful evaluation of our manuscript. The feedback provided has been extremely helpful in improving the clarity, scientific rigor, and overall quality of the work. We have carefully addressed all comments point by point, and detailed responses, including the corresponding revisions made to the manuscript, are provided below.
Responses to Reviewer 3
Comments 1: If natural deep eutectic solvents (NADES) are phytochemical extraction performance enhancers for the development of pharmaceutical and nutraceutical products, where is this development in research? To which plant was it applied, and what type of target metabolite was sought in the plant?
Response 1: The introduction has been expanded to to more comprehensively address the issues raised, providing additional context and background as recommended.
Comments 2: Why do they talk about antioxidant and antimicrobial activity in the title? Why do they talk about antioxidant and antimicrobial activity in the title? The manuscript also includes antifungal activity.
Response 2: We have used the term 'antimicrobial' to cover both antibacterial and antifungal activity. However, if the Reviewer recommends distinguishing these two groups more explicitly in the manuscript title, we would be happy to revise it accordingly.
Comments 3: Why test the antioxidant activity of phenolic compounds that are known to have this property, as stated on line 130 of page 4? The observed effect can be explained by the high antioxidant activity of thymol, which has been reported in numerous studies [22–26].
Comments 4: Why test the antimicrobial activity of compounds that are known to have that property, as stated on line 155 of page 7? Numerous studies have demonstrated the efficacy of pure thymol as an antimicrobial agent against Gram-positive bacteria (S. aureus, S. epidermidis, E. faecalis, B. cereus), Gram-negative bacteria (E. coli, S. Typhimurium, P. aeruginosa), and yeast (C. albicans) [27–29].
Response 3 and 4: We agree with the Reviewer that the antioxidant and antimicrobial activities of phenolic compounds and thymol are well-known and have been widely published. However, NADESs cannot simply be considered ordinary mixtures of compounds that exhibit specific biological activity when used separately. NADESs possess a supramolecular structure stabilized by hydrogen bonds, and it is these interactions that confer physicochemical properties distinct from those of their individual components. This raises the important question of whether their biological activity also differs from that of the pure constituents. Addressing this issue is essential for determining whether the properties of NADESs align with the principles of green chemistry and must be carefully considered when designing scientific experiments involving these solvents.
Comments 5: I think I should consider giving another twist to the research presented.
Response 5: While we respect the Reviewer's opinion regarding the context of our research, we believe that this manuscript addresses the limited knowledge surrounding the biological activities of NADESs. The data contained within may be useful for selecting extractants in the pharmaceutical and food industries, as well as for further scientific research.
Reviewer 4 Report
Comments and Suggestions for Authors
Lines 55-56 Please check if the words oxida-tion and exhib-iting are written correctly.
Line 143 Figure 1. The color of the Phenolic acid-base NADES in the caption is different from that observed in the figure, please check.
Lines 157 and 390 Write the species in lowercase and italics.
Line 184 Figure 2. The names of the samples are barely visible. The image should be improved.
Line 384. Mention whether an authorized biosafety protocol was used to handle pathogenic strains.
Author Response
We would like to express our sincere gratitude to all reviewers for their valuable time, insightful comments, and constructive suggestions. We truly appreciate the effort invested in the careful evaluation of our manuscript. The feedback provided has been extremely helpful in improving the clarity, scientific rigor, and overall quality of the work. We have carefully addressed all comments point by point, and detailed responses, including the corresponding revisions made to the manuscript, are provided below.
Question 1
Lines 55-56 Please check if the words oxide-tion and exhibit-ting are written correctly.
Answer:
We thank the Reviewer for noticing this mistake. The misspelled word has been corrected in the revised version of the manuscript.
Question 2
Line 143 Figure 1. The color of the Phenolic acid-based NADES in the caption is different from that observed in the figure, please check.
Answer:
We thank the Reviewer for noticing this inconsistency. The colors in Figure 1. have been corrected to ensure full consistency between the figure and its caption. The updated version now clearly represents the correct color coding for all NADES groups.
Question 3
Lines 157 and 390 Write the species in lowercase and italics.
Answer:
We thank the Reviewer for this comment. We have assumed this suggestion refers to Salmonella Typhimurium, as the rest of the species are in lowercase italics. According to the International Code of Nomenclature of Prokaryotes (ICNP), genus and species names are Latin binomials and are therefore written in italics, with the genus capitalized (e.g. Salmonella enterica). In contrast, “Typhimurium” is not a species name but a serovar designation. Serovar names are written in roman (upright) type and capitalized, not italicized. Thus, the taxonomically correct form is Salmonella enterica subsp. enterica serovar Typhimurium.
Question 4
Line 184 Figure 2. The names of the samples are barely visible. The image should be improved.
Answer:
We appreciate the Reviewer’s comment. The figure has been updated to improve visibility—sample names have been enlarged.
Question 5
Line 384. Mention whether an authorized biosafety protocol was used to handle pathogenic strains.
Answer:
Thank you for your valuable comment. We would like to clarify that all experiments involving pathogenic microorganisms were conducted in strict accordance with the applicable institutional biosafety regulations. Specifically, work involving bacterial and fungal pathogens, including methicillin-resistant Staphylococcus aureus (MRSA) and Candida auris, was performed in a biosafety level 3 (BSL-3) laboratory at the Department of Pharmaceutical Microbiology, Medical University of Lublin. All procedures were conducted under an authorized biosafety protocol approved by the Institutional Biosafety Committee.
We have added this information to the revised manuscript (Methods section 3.3.2 “All experiments involving pathogenic microorganisms were conducted in accordance with institutional biosafety regulations, in a biosafety level 3 (BSL-3) labora-tory at the Department of Pharmaceutical Microbiology, Medical University of Lublin, under a biosafety protocol approved by the Institutional Biosafety Committee.”).
Round 2
Reviewer 1 Report
Comments and Suggestions for Authors
Since authors have incorporated all suggestions, i would recommend this manuscriptipt for publication.
Author Response
Dear Reviewer,
Thank you for your comments, which have enabled us to improve the manuscript.
Reviewer 2 Report
Comments and Suggestions for Authors
This study evaluates the antioxidant and antimicrobial activity of 40 deep natural eutectic solvents (NADES) based on primary (PRIM) and specialized (HEVO) metabolites. The results demonstrate that lipophilic NADES containing thymol (HEVO) possess potent antioxidant and antibacterial activity, especially against resistant strains such as Staphylococcus aureus (MRSA), in some cases outperforming conventional antibiotics. The work provides solid evidence of the potential of these green systems as alternatives in pharmaceutical and food applications, supporting their rational use in extractions and formulations with intrinsic biological activity.
The only observations are: "separate '±' from the mean and the value of the variation," and the compound names are capitalized at the beginning of sentences.
Author Response
Dear Reviewer,
Thank you for your comments, which have enabled us to improve the manuscript. We hope you are satisfied with the corrections.
Reviewer 3 Report
Comments and Suggestions for Authors
I believe it's incorrect to use abbreviations in the title.
You should change the keyword "antimicrobial activity" since you refer to antimicrobial activity, just like in the title; it's repetitive.
You talk about solvents, but they are formulations.
You formulate 40 NADES, but you don't describe the properties of each compound. I believe you should include the chemical structure and provide more information.
You create formulations that are left at a temperature of 60 degrees for half an hour and then stored for a week. Some precipitate. You perform spectrometric or spectroscopic tests to confirm the identity and the properties you seek to form in the proposed NADES, such as: Polarized optical microscopy (POM), thermal analysis, and viscosity measurements?.
Why do they talk about Figure 1 and then Figure 5 in section 2.1? There should be a chronological order?
You should check your writing and spelling. For example, on page 10, you write: Salmonella Typhimurium, or Gram-negative bacteria (E. coli, S. Typhimurium, P. aeruginosa).
Review the figure captions for all figures, abbreviate the names of the microorganisms, and correct any misspelled ones.
Review all names of microorganisms and abbreviate them if they have already been mentioned in the text.
Some references do not have a DOI.
Author Response
Dear Reviewer,
We are very grateful for your critical assessment of our manuscript and your constructive comments, which have enabled us to enhance its quality. We have removed two abbreviations from the title. We have also removed 'antimicrobial activity' from the keywords. The reference to Fig. 5 has been removed from section 2.1. The full names of microorganisms are now used only when they are first mentioned, after which the appropriate abbreviations are used. We would also like to point out that the current spelling of 'Typhimurium' is in plain text and begins with a capital letter. Where possible, we have added the DOI number (one source does not have a DOI number).
Regarding chemical formulas, we believe that republishing them would unnecessarily increase the volume of the manuscript. These structures are well known, and their biological activity is described in numerous scientific publications. The physicochemical properties of NADES, however, are a topic for a separate paper. A significant number of publications already discuss the properties of NADES, especially those based on PRIM. We are currently compiling data on the properties of the benzyl alcohol:thymol system, including DSC, FTIR, NMR analysis and viscosity measurements. This system alone merits a separate publication.
We appreciate your understanding of our research approach. We would prefer our findings not to be lost amongst the vast amount of data describing the biological and physicochemical properties of NADES.